# Playing for a Resilient Future: A Serious Game Designed to Explore and Understand the Complexity of the Interaction among Climate Change, Disaster Risk, and Urban Development

**DOI:** 10.3390/ijerph18178949

**Published:** 2021-08-25

**Authors:** Wei Gao, Yuwei Guo, Fanying Jiang

**Affiliations:** College of Forestry and Landscape Architecture, South China Agricultural University, Guangzhou 510642, China; gaowei@scau.edu.cn (W.G.); yueiguo@163.com (Y.G.)

**Keywords:** serious game, disaster risk reduction, decision making, trade-off, urbanization, urban resilience

## Abstract

Urban development and disaster risk are deeply linked, especially now when we are facing increasingly frequent climate change. Hence, knowledge of the potential trade-offs between urban development and disaster risk reduction (DRR) may have potential to build a resilient and sustainable future. The objectives of this study are (1) to present education for a sustainability (EfS) program and to evaluate its performance: a serious game of knowledge communication for the interactions among climate change, disaster risk, and urban development; (2) to explore factors that will influence the players’ decision making in the trade-offs between urban development and DRR under an urbanization background through counterfactual scenarios constructed by a series of serious games. The Yudai Trench, once a critical component of the urban green infrastructure of ancient Guangzhou, has disappeared under rapid urban expansion, leaving the city exposed to environmental hazards caused by climate change. Is the disappearance of the Yudai Trench an inevitable event in the progress of urbanization? To answer this question, the study constructed counterfactual scenarios by recuring the historical progress through the same serious game. Gameplay involved the players’ decision making with associated impacts on the urbanization progress and the DRR in diverse climate hazard scenarios. For this study, 107 undergraduates from related majors, who are also would-be policymakers, were selected as players. The methodology combined questionnaire survey and participant observation complemented by interviews. The *t*-test results indicated that undergraduates’ knowledge levels had significant positive changes after the end of the serious game. Importantly, the results showed that the knowledge could potentially contribute to the players’ decision-making process for DRR by assisting them in making pre-decision. Beside this knowledge, the results expanded the range of influencing factors and solutions reported by previous literature on DRR under an urbanization background against climate hazards by constructing counterfactual scenarios, e.g., higher economic levels and policy incentives. In this study, the serious game was evaluated as an innovative communication and the EfS method in counterfactual scenarios. These findings of the study provide a reference for future practice, policymaking, and decision making so as to help harness lessons learned from unrealized environmental hazards to support a more resilient future through informed policies and plans.

## 1. Introduction

Urbanization is a common development trend in the world and a sign of human civilization and progress. However, as cities and clusters of cities, or conurbations, expand, the higher infrastructure strain and the increasing energy demand have led to ecological degradation [1]. More frequent climate change has accelerated the generation of hazards and vulnerability underlying the impact of urban growth [2], which has led to climate change-related extreme events that have presented a great challenge to disaster risk reduction (DRR) policy and management [3].

Previous studies on the relationships among urban development, climate change, and disaster risk are extensive.

The complex and intertwined interaction between urban development and disaster risk has been illustrated in numerous studies: rapid urban development increases disaster risk and vulnerability, and disaster risk often leads to significant losses and hinders further urban development [4]. Frequent climate disasters have caused huge economic losses and human casualties over the past twenty years, while the frequency of climate disasters is associated with unsustainable large-scale land development, which increases vulnerability and reduces resilience through deforestation, wetland reclamation, and urban expansion, etc.

For the relationship between climate change and disaster risk, White et al. [5] argued that disaster risk is increasingly understood as embedded in the urban development process and linked to the goal of limiting climate change. Dwirahmadi et al. [3] pointed out the importance of DRR in the face of frequent climate change, which may help people learn from previous disaster events to cope with future climate disasters and increase their coping capacity. In order to build better urban resilience, some studies recommend DRR as an important element of a governance framework to cope with climate hazards [6,7].

Identifying the resilience building as one of the central concepts, the 2030 Agenda for Sustainable Development incorporates goals related to DRR into several Sustainable Development Goals (SDGs) [8]. Furthermore, the Sendai Framework for Disaster Risk Reduction 2015–2030, a global framework for damage and risk reduction, emphasizes the need to focus on factors that increase disaster losses and risks and to address them [4,9]. However, despite more attention, DRR-related studies and practice have been increasing in recent years, but they are mostly focused on reducing disaster losses and risks through post-disaster response and management. The understanding of the potential drivers of disaster risk remains inadequate. This knowledge gap may be one of the reasons why the risks and losses associated with disasters are still increasing. Hence, a greater understanding of these risk drivers in the context of climate change is essential for sustainable risk management and policies under the urbanization background.

A possible view for this question is to consider the trade-offs between urban development and DRR processes in the context of climate change. Trade-offs, defined as choices involving diminishing or giving up one factor or fulfilling outcome in return for gaining another, have been explored in a wide variety of fields [4]: finance [10], land-use planning [11,12,13], and business [14]. The trade-offs between urban development and DRR are involved in many aspects such as community operation [3,15,16,17] and policy intervention [18,19]. In either case, the trade-offs in risk management and policies need to be made or assisted by people with the relevant knowledge, and the trade-off thinking of these professionally educated professionals is likely to influence the final outcome. Hence, articulating factors that may influence the trade-off thinking of professionally educated professionals or would-be professionals may lead to a better understanding of the obstacles and blind spots in existing DRR practices, thus helping professionals, or would-be professionals, to better navigate both urban development and climate DRR processes in the context of climate change.

Observing and recording the risk decision-making process in different scenarios may be a good way to visualize and analyse the trade-offs between urban development and DRR. Serious games, generally involving the use of scenarios for players to make decisions [20], are considered a participatory method and are used to understand the stakeholders’ reactions to different scenarios, and to teach them the impact of adaptation [21]. Many studies in recent years have proven that serious games are an effective tool for environmental education or EfS in many fields such as such as water management [22], marine conservation [23], and urban planning, environmental design, etc. Most notable is the series of studies conducted by Alenka Poplin and her team exploring virtual campus environment design [24], virtual urban market design [25], etc. Skinner et al. [26] convened a session of “Games for Geoscience” at EGU General Assembly, in which games were used and as a method to train the public, practitioners, and decision makers in order to build environmental resilience. Serious games are used to explore the concerns of city managers, citizens, and various stakeholders [27], and simulate negotiation and consensus building in public engagement sessions [28]; other directions have had extremely good results. As a serious game for research, in addition to creating browser versions of online games, game design can be done with the help of open game platforms to achieve better results, such as an urban design game based on Minecraft [29], which involves children in Brazil in urban design decisions; such as the SimTorino project based on the SimCity initiative, which simulates urban development and disaster risk response in the next twenty years [30]. However, there is room to explore the relevant research in serious games in terms of exploring the relationship between urban development and DRR based on historical scenarios and using the perspective of counterfactual scenario analysis.

Against this backdrop, selecting undergraduates in related majors, who are the would-be professionals, as typical and representative subjects, this study aimed to present a serious game to analyze the players’ decision making for trade-offs in development and DRR in the context of climate change. The main objectives of this study are (1) to propose an SG and evaluate its performance in climate disaster-related EfS for students who have received professional education; (2) simulating different scenarios in urbanization progress, climate hazards, and historical and policy scenarios in the SG to understand how DRR decisions are made and change with shifts in the process with factors such as an individual’s risk knowledge and economic and policy incentives.

By reorganizing and understanding the trade-offs in would-be professionals’ decision making in the context of climate change, this study attempts to provide a creative perspective for improving the relationship between risk-generating development and disaster risk in the current climate change context, as a way to integrate DRR and resilience for sustainable development.

## 2. Materials and Methods

### 2.1. Study Site

The serious game was set using the history of the Yudai Trench. Once located in Guangzhou, China, and dug during the Song Dynasty in 1011, the Yudai Trench initially functioned as a safe harbor. Since its excavation, the Yudai Trench, which is more than 60 m wide, gradually developed functions, such as water supply, drainage, and flood resistance, making outstanding contributions to climate disaster risk reduction and urban resilience in the ancient city of Guangzhou [31]. However, as urban construction accelerated and the surrounding neighborhoods continued to expand, Yudai Trench began to fill up with silt. Yudai Trench was finally converted into an underground channel due to ineffective dredging in 1952 [32].

As an important green infrastructure, Yudai Trench’s disappearance means that climate DRR has given way to urban development. In fact, the thousands of years of the rise and fall of Yudai Trench is a constant process of trade-off between urban development and climate DRR.

Through counterfactual scenarios in the game, the study restored the width of the Yudai Trench when it was first excavated, allowing players to redo the trade-offs between urban development and climate DRR, and to rebuild the urban resilience. The 20-round game, which was a sequential process defined by several scenarios based on historical records about the socioecological system evolution of the Yudai Trench over thousands of years. Based on the historical data, the study selected 8 historically recorded climate disasters in Guangzhou as the climate hazard scenarios for the game. In addition, these scenarios also consisted of varying policy and historical contexts representing the urbanization progress (Table 1). These scenarios would appear sequentially in the game session. The game rules and the scenarios with sufficient historical evidence ensured the reality and seriousness of the game [33].

While combing through the historical data, we discovered that the long historical span of the Yudai Trench meant it had gone through several historical periods with different economic levels and different policy preferences: according to historical records, the coast of Yudai Trench was only a commodity market when it was first dug, then gradually developed into the most prosperous financial and commercial center of Guangzhou, and then gradually declined at the economic level after being affected by the blockage of the Yudai Trench and wars, etc. At the same time, according to historical records, the government organized the dredging of the Yudai Trench by implementing policies.

That is, based on the compilation of historical data, economic levels and policy preferences may be factors influencing the trade-offs affecting urban development and climate DRR that existed in the historical process of the Yudai Trench. Furthermore, Lebel et al. [34] considered that risk decision making is influenced by an individual’s risk knowledge.

Therefore, this study focused on the effects of knowledge, economic level, and policy incentives on in-game decision making representing players’ trade-offs.

### 2.2. The Game Settings

At the beginning of the game, players have some initial funding and are told that there will be some climate disasters coming up in the game. Players can use the diverse cards to optimize development and climate DRR in the context of climate change or to achieve a balance between the two (Table 2).

Each round of the game consisted of the following three steps:The game moderator announced the specific game scenario for that turn (as the number of turns increased, the width of the Yudai Trench narrowed, and the frequency and intensity of climate disasters increased);Each player in the game session advanced on the game board in turn by rolling the dice. Players can buy arrival plots and construct on them for development or climate DRR;Each player in turn reckoned the gains and losses made during the turn.

By constructing counterfactual situations, the game allowed players to make trade-offs and decisions freely. Over the course of the game, the overall resilience decreased as the width of the Yudai Trench narrowed, which induced more climate disasters. Players can slow or reverse the increasing level of silt in the Yudai Trench by purchasing and placing water cards on river plots, or they can build farmland and forest to improve the overall resilience of the site in another way. Of course, players can also focus on city development, population expansion, and store construction.

Ultimately, the winner in each game session was the player who had established better balance with the most hazards resistance and profit. While none of players knew the winning criteria until the end of the game, which was consistent with the reality that no one would tell you exactly what you should do to be right, it all depended on the player’s own judgment.

### 2.3. Participants and Game Sessions

One hundred and eight junior to senior undergraduates, who majored in landscape architecture from 6 classes in Guangzhou, China, participated in the serious game (Figure 1). The maximum number of players that could be included in each game session was 6. All players participated voluntarily by responding to an open invitation. One hundred and seven of them were recorded, where each player was the source of the data or observations that were analyzed. In this process, one of them did not participate in the pos*t*-test questionnaire after the game for personal reasons, so we did not accept his feedback as research data. The participants’ mean age was M = 21.7 (SD = ±1.88), and the gender distribution was even (56.6% female).

From October 2020 to March 2021, 18 game sessions were organized. These game sessions were equally assigned to three experimental groups corresponding to different treatments: the control group (T1, *n* = 36); the group provided with risk knowledge about the damages brought by climate hazards and how to resist them in the game (T2, *n* = 36); and the group with three times the initial funding than the other two (T3, *n* = 35).

In pretests, we found that the students’ attention gradually faded after approximately 15 rounds. When the game went beyond twenty rounds, most players started to get impatient. To maintain players’ concentration, the game was compressed to 20 rounds. Each round took about 5 min to play on average.

### 2.4. Data Collection Methods

To demonstrate the effectiveness of the serious game as an Efs Method, questionnaires and interviews were used to validate the educational effect of the game. In our study, the questionnaire method was mainly used to collect data for quantitative analysis, while the information and data obtained from the interviews were mainly used to corroborate the results of the quantitative analysis and to further understand the players’ attitudes, which were used to conduct qualitative analysis. This combination of quantitative and qualitative data collection methods is often used to demonstrate the validity of Efs, as in the studies by Orduña et al. [35] and Mahmud et al. [36].

In the collection of players’ decision-making outcomes, participatory data collection methods such as interviews, workshops, and in-game observations are considered effective heuristics [37,38]. Participatory data collection methods have several advantages, including the ability to record and analyze every decision made by different stakeholders [35]. Therefore, scenario-based analysis of the decision-making process generated by using participatory data collection methods as a basis is considered to be highly meaningful [39]. We therefore used participant observation in this process, supplemented by interviews.
The questionnaire survey: consisted of a scale and a written feedback form (Appendix A). The scale measured knowledge (8 items) on the causes and impacts of variable climate hazards and the interaction between climate hazards and urbanization. Twelve items were originally developed with reference to the cognitive learning objectives for climate action in Education for Sustainable Development Goals: Learning Objectives [40]. Kieser et al. [41] argued that the recommended minimum sample size for trial test is 20–40 when the total experimental sample is in the range of 80–250, and this was further argued in a subsequent study by Hertzog [42]. Therefore, a total of 22 junior and senior undergraduates were tested in the trial test. We excluded the four items with low total correlation. The five-point Likert scale is often used to explore the effects of climate change or climate disasters, such as in Shen et al.’s study, where a five-point Likert scale was used to describe residents’ perceptions of the effects of typhoons [43], and in Khoury et al.’s study, where a five-point Likert scale was used to describe participants’ perceptions of floods [44]. The five-point Likert scale was considered a valid tool for making degree judgments and attitude expressions in the above studies, while scales with too many options were considered to have the potential to confuse respondents [45]. Thus, we used a five-point Likert scale ranging from 1 to 5, where 5 indicated completely understand and 1 indicated do not understand at all. The pretest data were collected two weeks before the game. The Cronbach’s α reliability (pos*t*-test) of the knowledge scale was 0.81, and the written feedback form enabled players to express their subjective feelings of the game. Players were asked to give feedback on their first thoughts and emotions of the game and to evaluate the learning effect in the game. Players were asked to complete the pos*t*-test data and the feedback form.The in-game observations: because of the large number of observations, three observers, who were responsible for keeping track of different things, were assigned to every game session. The observers consisted of postgraduates, who were asked to record the player’s decisions and strategies without interfering with them. The recorded decisions included the overall performance, process, and results of players’ decision making (Table 3).The follow-up interview: the question posed to the students covered strategies used in the game to increase resilience; considerations when making certain decisions; their understanding of the interaction between urbanization and climate hazards; and their feelings during the risk decision-making progress.

We used six indicator variables to describe different aspects of the players’ decision making, which were recorded during the in-game observations. First were two measures of the overall resilience performance: Climate DRR and Efficiency in Climate DRR.

Because the gameplay did not fully cover the dimensions of the typology framework proposed by Tuhkanen et al. [4], we chose four indicator variables, representing the three key dimensions in the trade-offs framework: aggregation, time, and participation, to measure the players’ decision making about trade-offs between development and DRR. These indicator variables are: Green Infrastructure Development, Economic Development, Timeliness, and Improvement.
Green Infrastructure Development and Economic Development in the aggregation dimension: the aggregation dimension incorporates the conflict between desired planned economic growth and the damage to social and environmental well-being that comes with economic growth [4]. During the in-game observations, the observers were asked to record the numbers of social cards and ecological cards used in each round to evaluate the balance and conflicts between economic development and climate hazard risk reduction.Timeliness in the time dimension: the time dimension tackles the conflict between potential long-term disaster risk and short-term high-return gains [4,46,47]. Before the game started, all players were told that there will be some climate hazards in the game, but all of them did not know in which round the hazards would appear. The observers were asked to record the number of rounds in which the player started building green infrastructure with ecological cards to evaluate the players’ trade-offs in decision making between the short-term and long-term gains.Improvement in participation dimension: the participation dimension involves the stakeholders’ sharing of power, communication, and cooperation in the decision-making process [4]. In the gameplay, the hazard-related decision-making process of each player in the game session took place in the same game board, where players operating their own territories negotiated their perceived trade-offs based on their competing stakes. However, to maintain a sufficient width of the Yudai Trench, stakeholders needed to cooperate and identify shared values through which shared benefits are created. It means that each player needed to be aware of the interdependence between players during the trade-off, and the results of the trade-off can be visually identified by comparing whether the width of the water system at the end of the game has improved along with the real width.

Overall, effectiveness of the SG as an EfS method was evaluated by (1) the improvement of the players’ knowledge level by the end of the game; (2) requiring the players to judge the SG based on the theme, length, game strategy, ease, entertainment, and balance of mechanics [35]. In addition, the data and information recorded during the in-game observations and the follow-up qualitative interviews will be used to measure the players’ decision making about trade-offs between development and DRR.

### 2.5. Analysis Methods

All statistical analyses of the quantitative data were performed using SPSS. In our study, we used *t*-tests to analyze the quantitative data.

In our study, we chose to use paired *t*-tests to evaluate whether there is a significant difference between the pre-test and pos*t*-test results of participants’ knowledge scale of the same group. In the analysis of in-game observations, we used independent *t*-tests for data analysis as a way to determine whether there are significant differences in the recorded data generated by players in groups with different control conditions during game play. This is because independent *t*-tests are applied to mean comparisons between two independent groups, while for paired data, paired *t*-tests can be performed [48].

Essentially, the *t*-test allows us to compare the means of two data sets and establishes the problem statement by assuming a null hypothesis that the two means are equal as a way to determine if there is a significant difference between the two data sets. Therefore, if the *t*-test values show that the null hypothesis qualifies to be rejected (*p* < 0.05 at the 95% confidence level; *p* < 0.01 at the 99% confidence level) [49], it means that there is a significant difference between the means of two groups.

## 3. Results

### 3.1. Evaluation of the SG as an Education for Sustainability Method

We used paired *t*-tests to analyze the pre-test and pos*t*-test data of the three groups of players before and after the intervention, respectively. The three experimental groups corresponded to different treatments: T1 was the control group (n = 36); T2 was the group provided with risk knowledge about the damages brought by climate hazards and how to resist them in the game (n = 36); T3 was the group with three times the initial funding than the other two (n = 35). The average scores for knowledge for the three groups before and after the serious game are listed in Table 4. The *t*-test results indicate that the knowledge levels of all of the students were significantly enhanced by the end of the serious game (*p* < 0.01 implies a statistically significant difference between the two sets of paired data within the 99% confidence level.).

The player feedback on the SG is present in Figure 2. Players found the game easy to understand and had many chances to win. According to over 90% of players, the SG created an interesting environment for open discussion and analysis of a wide variety of topics. Particularly, players who were less familiar with climate hazards enjoyed the SG as a way to popularize climate and environmental issues. In general, the length of the game sessions was acceptable, while some players felt that the game was too short to show the advantages they had accumulated.

The game sessions improved knowledge by allowing players to experience conflicting social, economic, and environmental values in a socioecological system through counterfactual scenarios. Over 91% of players confirmed that they gained new knowledge by playing the SG, 6% were unsure if they learned anything new, and 2% thought they gained no knowledge in the game. We learnt that even the professionally educated players generated new relevant knowledge about climate hazards during the SG. The relevant feedback from players was as follows:
*“It is effective to deliver the knowledge about climate change via games, especially for those who dig in this topic for the first time.”*
*“It was an interesting experience. I got to know some knowledge about city resilience via an intuitional and visual method, such as how to prevent some risks or disasters effectively by strengthening green infrastructure construction.”*
*“The contents of the game are actually closely related to our daily life, which made me feel like thinking of strategies for real practice. Besides, I learnt that the implementation of city resilience depends on many factors, one of which is the reasonable configuration of land resources.”*
*“For the first time I got to know we can learn from games. I think the topic implied by this game is of practice significance. I hope I can apply the knowledge that I learnt from this game in my real life someday.”*

Based on the feedback in the questionnaire survey and interviews, we concluded that players increased their knowledge of resiliency of socioecological systems, as well as the types, causes, and impacts of variable climate hazards underlying an urbanization background. More importantly, this program inspired the players in that they could link EfS with gamification design and viewed serious games as a promising method to educate and learn about socioecological decision making.

### 3.2. Analysis of Decision Making between Development and Climate DRR

#### 3.2.1. Risk Knowledge

The decisions and results over the course of the gameplay were collected to explore how risk knowledge about the damages of climate hazards and ways to resist them influences decision making. The results in the context of climate hazards (i.e., drought, typhoon, and flood) are presented in Table A1 in Appendix B, which indicate that players with risk knowledge built a better resilience than players in T1, who were without risk knowledge, as evidenced by the better score in Climate DRR (T1: 0.44 ± 0.16, T2: 0.71 ± 0.15, *p* = 0.001 < 0.01) and Effectiveness in Climate DRR at the end of the game (T1: 5.67 ± 1.53, T2: 7.44 ± 1.48, *p* = 0.000 < 0.01). Meanwhile, players with risk knowledge showed better Timeliness in Climate DRR of risk decision making in hazard response (T1: 4.30 ± 1.22, T2: 1.67 ± 1.29, *p* = 0.001 < 0.01) and a better balance between Green Infrastructure Development and Economic Development (Figure 3).

As displayed in Figure 3, with common understanding of the presence of climate hazards in the game, players in T1 lagged significantly behind players in T2 in terms of the Timeliness in Climate DRR: players with risk knowledge, on average, started building green infrastructure at the beginning of the game, while players without risk knowledge did not do so until the first climate hazard was coming. Decision making in advance of disasters guided by risk knowledge may be one of the reasons for the difference in DRR between T1 and T2. In addition, the lack of risk knowledge may lead decision makers to ignore the risk of hazards [5,50], which made them more concerned about the level of economic development before they were really aware of the damage of disasters.

Furthermore, there was a significant lag between starting to build green infrastructure and being able to resist climate hazards (Effectiveness in Climate DRR) for players without a risk knowledge (Figure 3a), which was not evident in players with risk knowledge. In the qualitative interviews, many players in the T1 group said that it took them a long time to figure out how to resist the climate hazards in the game: *“It took me five to six attempts before I got the right way to resist a flood, but I had lost a lot of profit before that.”* It may suggest the importance of risk knowledge in accurate and timely DRR action [51].

Additionally, despite the statistically significant differences on the level of Effectiveness in Climate DRR between T1 and T2 by the end of the game, it was found that the gap narrowed significantly in the later stage of the game (Figure 3). It may depend on the player’s self-learning when faced with the climate hazards [34,52]. This view is well supported by the results of the qualitative interviews, with some in T1 saying they prefer the latter stages of the game to the earlier stages, because at that time, they found they can understand how to play and make better decisions according to the understanding of the impact of climate hazards: “*At first, I didn’t know what the implications of the hazards would be. After playing for a while, I felt I knew how to play*”.

#### 3.2.2. Economic Development Level

Table A1 in Appendix B also lists the average scores of each variable for players in T3, revealing the impact of economic development level on decision making between urban development and climate DRR. The Climate DRR level of players in T3 was significantly higher (T1: 0.44 ± 0.16, T3: 0.56 ± 0.11, *p* = 0.001 < 0.01), which indicates that with the improvement in the economic development level, players devoted greater resources to safety, including implementing precautionary measures designed to reduce the impacts of climate hazards. As displayed in Figure 3, compared with the T1, T3 built more green infrastructure in the game to improve the resilience (T1: 2.38 ± 0.40, T3: 3.06 ± 0.33, *p* = 0.001 < 0.01). This concurs well with the previous findings [53,54].

Surprisingly, no significant difference was found between players in T1 and T3 regarding Timeliness in Climate DRR (T1: 4.30 ± 1.22, T3: 4.06 ± 0.94, *p* = 0.339 > 0.05). Figure 4 shows that before the first hazard, the majority of T3 players’ decision making still revolved around economic development rather than green infrastructure development. This proportion shifted significantly after the first hazard. In follow-up interviews, regardless of economic development level, many players said that they were more focused on making profit before they really realized that hazards could be so harmful to them.

Furthermore, the *t*-test results indicated that the Efficiency in Climate DRR at the end of the game increased significantly with the improvement in the economic development level (T1: 5.67 ± 1.53, T3: 7.00 ± 1.46, *p* = 0.001 < 0.01). This was probably due to the security sense brought by the ease of availability of funds, which makes players more willing to try. As in the follow-up interviews, some players in T3 expressed a willingness to try different strategic decisions to resist the hazards without being overly cautious, as the extra income significantly reduced their trial-and-error costs.

#### 3.2.3. Policy Incentives

Policy incentives related to urban resilience can also have an impact on players’ decision making. As displayed in Figure 5, there was a significant increase in Green Infrastructure Development during the period when the policy incentives were in effect, showing a significant policy response. The responsiveness may continue for some time after the end of the policy, which was especially evident for players in T1. In contrast, players in T2 were also affected by the policy, but in a relatively minor way.

Policy incentives related to urban resilience can also have an impact on players’ decisions making. As displayed in Figure 5, there was a significant increase in Green Infrastructure Development during the period when the policy incentives were in effect, showing a significant policy response. The responsiveness may continue for some time after the end of the policy, which was especially evident for players in T1. In contrast, players in T2 were also affected by the policy, but in a relatively minor way.

#### 3.2.4. The Overall Decision-Making Strategies Taken by Players during the Game

The strategies represent a series of the decision making between development and climate DRR of players systematically and distinctly. Different developments of resilience strategies can be observed in various game sessions with different prerequisites. We categorized the strategies observed throughout the sessions into two main types: fast-growth fatalist and protectionist.

Fast-growth fatalist is defined in the game as a player who used more social cards than ecological cards, whose strategy was to compensate for the losses caused by climate hazards by engaging in more economic activities. The players spent almost all resources developing new stores and expanding their land, while limited or no ecological cards were collected. The players tended to have a lot of stores and residences to sustain. When impacted by hazards or when losing access to resources due to a broken capital chain, the player could no longer sustain all stores and residences, resulting in bankruptcy. Contrarily, protectionist strategies focused on risk reduction and resilience development. Players adopting such a strategy tended to develop their stores and expand their land slowly and used more ecological cards. Stores and residences therefore had a higher chance of surviving climate hazards and green infrastructures were more adapted to resist it.

We found that players in T2 and T3 preferred playing as protectionists, while players in T1 tended to play as fast-growth fatalists. Furthermore, when there were more protectionists than fast-growth fatalists in the game session, it was more likely that the Yudai Trench survived in the context of urban expansion, which was an improvement over the actual situation (Figure 6). However, are all game sessions in which there are more fast-growth fatalists than protectionists unable to gain improvement at the end of the game? The answer is no (Figure 6). That is, when the game was dominated by fast-growth fatalists, the game did not necessarily end in defeat. This is indeed an interesting finding. Because these fast-growth fatalists did accumulate wealth faster than the protectionists in the early stages, the accumulation of sufficient wealth in the early stages may help some of these fast-growth fatalists to counteract all the adverse effects of climate disasters in the later stages. These fast-growth fatalists may rely on luck, but they could also use the ability to collaborate with each other. However, we do not think this is a long-term solution because it is likely that when there are enough turns in the game, the unsustainable growth strategy will highlight its disadvantages.

The overall analysis of decision making between development and climate DRR indicated that risk knowledge, higher economic levels, and policy incentives can positively influence and guide players’ climate DRR decisions, which are reflected in the dimensions of aggregation, time, and participation in the trade-off framework.

## 4. Discussion

DRR is an important approach to respond to disasters caused by climate change and is essential to building urban resilience and achieving sustainable development. A better understanding of the trade-offs that exist with urban development and climate DRR is needed to find development strategies that promote climate disaster resilience while maintaining stable urban development. Hence, the main objectives of our study were to explore how risk knowledge, economic levels, and policy incentives influenced the decision-making process and to assess the effectiveness of serious games as an EfS method.

### 4.1. Risk Knowledge to Provide a Sustainable Perspective for Climate DRR Decision Making

Risk knowledge and climate DRR have proved to be closely linked, and these connections can help us to improve DRR [55,56]. Our findings provide good evidence of the positive impact of risk knowledge in climate DRR decisions: participants with risk knowledge performed significantly better in the dimensions of aggregation, time, and participation in the trade-off framework as well as the overall resilience performance than those without risk knowledge.

Furthermore, we found that risk knowledge provided professionals, as decision makers, with a sustainable perspective, which is used to guide climate DRR decisions. The findings show that in the time dimension, risk knowledge would guide professionals to take a long-term perspective and prepare in advance for unhappened or unknown climate hazards. In the aggregation dimension, adequate risk knowledge would help professionals clearly perceive the importance of environmental well-being, rather than focusing only on economic construction. For the participation dimension, risk knowledge would be effective in enhancing the degree of consensus among professionals representing different interests in DRR decision makings, to some extent, creating a basic common perception of the value of green infrastructure for sustainable development. The sustainable DRR decisions may be influenced by higher risk perception and more proactive disaster-preparedness behaviors resulting from risk knowledge [34,57,58] and may also be related to the values that professionals are taught in their professional education that value environmental protection and sustainable development.

Based on the understanding that risk knowledge provides sustainable perspective for decision making, further work might focus on some of the challenges related to knowledge production and implementation in the DRR in real situations, which is a simplified part of the game. For example, these may include how to systematically collect and share disaster-related facts, data, and information [47,59], and how to remove potential barriers between knowledge and DRR practice that prevent effective use of existing risk knowledge [5,60,61].

### 4.2. Higher Economic Levels to Create Prerequisites for Positive Climate DRR Decision Making

Our study revealed that the DRR decisions at higher economic levels did not appear to have significantly better performance on each of the trade-off dimensions, while the overall resilience performance was effectively improved. That is, higher economic levels were not the decisive factor affecting the development–DRR trade-offs but can, to some extent, translate into DRR decisions that favor resilience.

This finding confirms previous evidence that the critical underlying factor in any economy’s response to a natural disaster is its level of wealth [53]. A possible explanation of the improvement of the overall resilience performance is that increases in economy increase the demand for safety in the face of disaster risk [54], which makes players willing to devote more resources to DRR. In addition, another possible reason is that the increase in economic level provides the premise for more investment of resources into DRR. As previous studies have pointed out, economic pressure is one of the obstacles to translating the theoretical cognition of DRR into implementation [5].

However, why the higher economic levels failed to make a sufficient impact on the development–DRR trade-offs? The same evidence can be seen in the history of the Yudai Trench: it began to fill with silt during its economic heyday, as the surrounding inhabitants began to fill in the river in order to acquire a larger area of land for economic development. In the study, we found that with a high economic level as a prerequisite, knowledge of the disaster risk may be an important turning point for DRR decisions: most players began to make decisions about DRR only after experiencing losses from climate disasters. The Sendai Framework for Disaster Risk Reduction also emphasizes the importance of disaster risk cognition [9].

Based on these results, we believe that low-income people and the communities in which they congregate should be the focus of attention in climate DRR policy and resilience building, because they face greater vulnerability in both pre-disaster prevention and post-disaster response due to their low economic levels. At the same time, we argue that economic level is not the only important development measure in disaster risk reduction and that higher levels of cognition are equally important. Thus, in addition to more direct DRR efforts, long-term climate DRR policies may include efforts to improve education and further raise the overall economic level.

### 4.3. Targeted Policy Incentives to Improve Stakeholders’ Coherence in Climate DRR Decision Making

Our study revealed that policy incentives on climate DRR can go some way toward guiding and encouraging professionals to make relevant decisions. Although these professionals represented their own interests in the game, and the decision situations varied across experimental conditions, the motivation of stakeholders in different decision situations for climate DRR decision making under the relevant policy incentives showed a consistent upward trend. This finding confirms previous evidence of the importance of incentives in collective action and decision addressing environmental degradation issue [62,63,64].

However, this conclusion seems to be contrary to the historical development process of the Yudai Trench in reality. According to historical records, there have been several dredgings of the Yudai Trench under the organization and incentive of the government in order to maintain the width; however, the effect was not obvious. The continuous siltation of the Yudai Trench has further exposed the city to the risk of climate hazards. This may be due to the fact that most of the dredging policies at that time were only motivated by shipping and water supply requirements, and there was a serious lack of recognition about the value of Yudai Trench, an important green infrastructure for the city, in climate DRR. The cognitive limitation at the policy level made the policy incentives’ lack of DRR pertinent, which makes it impossible to spontaneously form behaviors and decisions favorable for climate DRR. A study by Weichselgartner et al. [50] also argues for the importance of targeted policies in climate DRR under the influence of knowledge.

Therefore, linking policy incentives and risk knowledge may be an effective way to further address the DRR issue. Policy incentives targeted at effectively improving DRR would help reconcile potential conflicts among stakeholders that exist in the trade-offs between development and DRR, while also better integrating multiple scales, different decision makers, various sources of knowledge, and different disciplines into disaster risk research.

### 4.4. Effectiveness of the Serious Games as an Education for Sustainability Method

Our study revealed that the SG presented in the study facilitated the players’ understanding of resiliency of socioecological systems, as well as the types, causes, and impacts of variable climate hazards with an urbanization background. In addition, players could adjust their decisions and strategies by learning about climate risks in the game’s scenarios. Learning in the game scenarios is actually a double-loop learning process, referring to a revisiting of assumptions and to questioning the governing variables themselves by means of a critical reexamination [65], which makes the impact of the game more profound and effective. While the results suggest that serious games were effective in improving players’ knowledge, this was with a relatively limited sample size, mainly from the undergraduate students. Further studies of the method with additional data, scenarios, and audiences are needed to assess the potential of SG with a wider public.

The results also show the potential of the SG as a data collection approach because it effectively identifies the dynamics of the decision-making process representing multiple dimensions about the trade-offs between urban development and climate DRR that exist under the influence of potential factors.

## 5. Conclusions

This study explored the effects of risk knowledge, economic level, and policy incentives on professionals’ climate DRR decision-making process by constructing counterfactual scenarios about the Yudai Trench with the help of a serious game. The results from questionnaire surveys, in-game observations, and interviews suggest that while all three factors favored DRR, the impact of risk knowledge was the most prominent and fundamental, with the effects reflected in the three dimensions in the trade-off framework and the overall resilience performance. In addition, neither economic level nor policy incentives could contribute to the development–DRR trade-off in isolation from risk knowledge. Moreover, the study demonstrates that the proposed serious game is an effective EfS method for enhancing professionals’ understanding of the complexity of climate hazards.

The results of this study can contribute to the students involved in this serious game, who are also would-be policymakers, so that they can be aware of the importance of risk knowledge and the fact that different methods of decision making in the face of climate hazards can have completely different consequences. The results of this study can also help university teaching staff and authorities to adapt their curricula and approaches to the requirements of the Sustainable Development Goals for building a more resilient future.

Limited by the cognitive limitations brought about by the context of the times, the demise of the Yudai Trench seemed inevitable in the past, but it should not be in the present. In conclusion, this study is configured as an exploratory study that opens the way for new research. Insights from the serious game, combined with other theoretical research and practical experiences on DRR, should help advance more systematic and comprehensive decision support to improve climate risk management and policy in the context of urbanization and ultimately develop realistic strategies to achieve a resilient and sustainable future.

Future research could include expanding the sample to include undergraduate students with degrees in other fields or at other educational levels, or even the general public. Therefore, comparing the results obtained in this paper with cross-cutting expertise needed to achieve the Sustainable Development Goals would enrich the literature.

## Figures and Tables

**Figure 1 ijerph-18-08949-f001:**
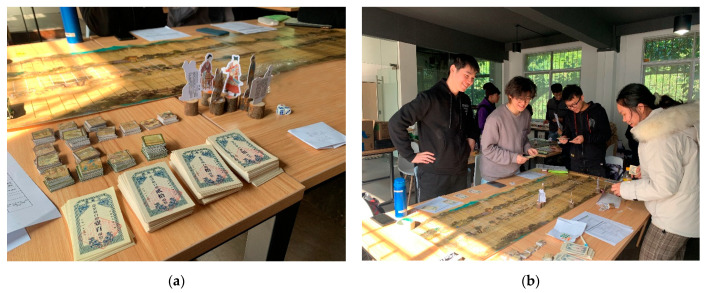
(**a**) The components of the serious game; (**b**) the game process.

**Figure 2 ijerph-18-08949-f002:**
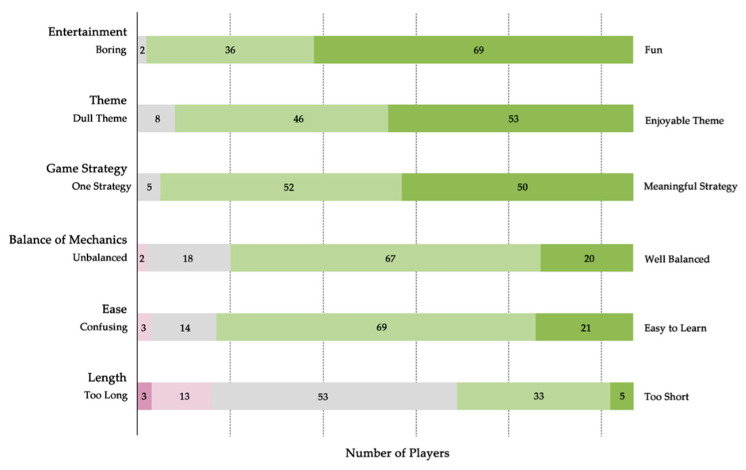
Feedback on the SG sessions.

**Figure 3 ijerph-18-08949-f003:**
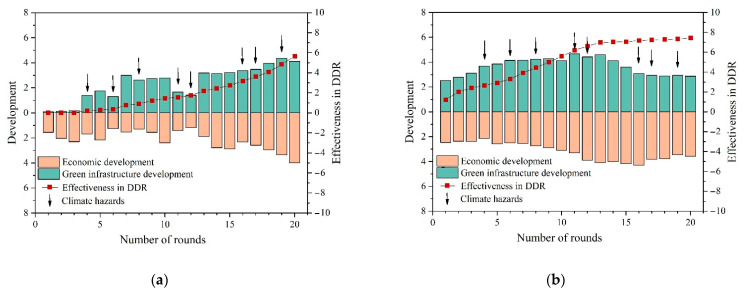
(**a**) Players’ decision making about trade-offs between development and DRR in T1; (**b**) players’ decision making about trade-offs between development and DRR in T2.

**Figure 4 ijerph-18-08949-f004:**
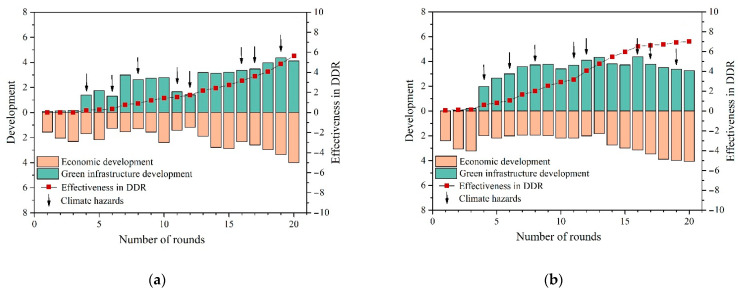
(**a**) Players’ decision making about trade-offs between development and DRR in T1; (**b**) players’ decision making about trade-offs between development and DRR in T3.

**Figure 5 ijerph-18-08949-f005:**
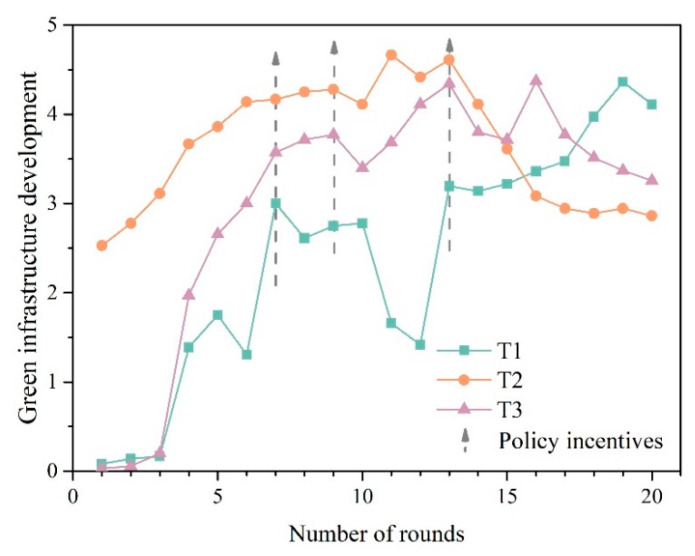
Players’ performance in Green Infrastructure Development with policy incentives.

**Figure 6 ijerph-18-08949-f006:**
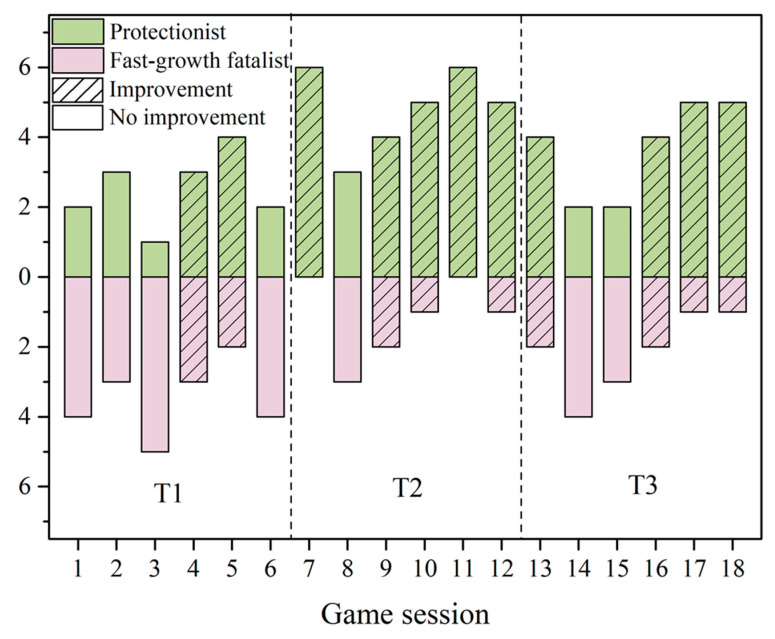
Strategies adopted by players in different game sessions and the Improvement by the end of the game.

**Table 1 ijerph-18-08949-t001:** Description of the SG’s scenarios.

Type	Scenario	Description in the Game
Urbanization Progress	Development: Song Dynasty (1011–1271) and Yuan Dynasty (1271–1380)	Wide Yudai Trench, slow climate change, and low-intensity climate hazards.
	Heyday: Ming Dynasty (1380–1644) and Qing Dynasty (1644–1918)	Narrowing Yudai Trench, accelerating climate change.
	Decline: The Republic of China (1918–1949) and People’s Republic of China (1949–the present day)	Clogged Yudai Trench and high-density climate hazards.
Climate Hazards	Drought	Population decline.
	Typhoon	The residences are damaged based on the typhoon intensity.
	Flood	The profit of stores is reduced based on the flood intensity.
Policy Incentives	Grain Demand	The Farmland is subsidized.
	Sedimentation Dredging	The cost of Water is reduced.
	Encourage Tree Planting	The cost of Forest is reduced.

**Table 2 ijerph-18-08949-t002:** The types of cards used for the development and DRR of the Yudai Trench.

Type	Sub-Type	Description
Ecological Card (for Climate DRR)	Farmland	The key to reducing damage from drought.
	Forest	The key to prevent a flood.
	Water	The sufficient width of the Yudai Trench is the basis for maintaining the stability of the socioecological systems, as well as the key to resisting typhoons.
Social Card (for Development)	Residence	Accommodating the resident population for development.
	Store	Gaining profits.

**Table 3 ijerph-18-08949-t003:** Data needed to record during the in-game observations.

Type/Dimension	Variables	Description
Basic Information	Game Session	1–18
	Rounds	1–20
	Scenarios	Urbanization progress, climate hazards, and historical and policy scenarios in each round.
	Players	1–107
Overall Resilience Performance	Climate DRR	(0–1). The percentage of plots that can resist all hazards by the end of the game for each player.
	Effectiveness in Climate DRR	(1–8). Number of hazards (occurred and not occurred) that each player can resist according to the decision-making results in each round.
Aggregation	Green Infrastructure Development	(1–10). Number of ecological cards used in each round for each player.
	Economic Development	(1–10). Number of social cards used in each round for each player.
Time	Timeliness in Climate DRR	(1–20). Number of rounds in which the player starts building green infrastructure with ecological cards.
Participation	Improvement	(Yes/No). Whether the width of Yudai Trench improved by the end of the game compared with the real situation.

**Table 4 ijerph-18-08949-t004:** The *t*-test scores on the effects of knowledge.

Treatment	Pretest		Post-Test		*p*-Value
	Mean	SD	Mean	SD	
T1	3.476	0.306	4.132	0.296	0.00 **
T2	3.462	0.324	4.188	0.389	0.00 **
T3	3.493	0.239	4.139	0.264	0.00 **

** *p* < 0.01.

## Data Availability

The data presented in this study are available in Table A1 in Appendix B.

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
