# Peer review of "Playing for a Resilient Future: A Serious Game Designed to Explore and Understand the Complexity of the Interaction among Climate Change, Disaster Risk, and Urban Development"

_ijerph, 2021, doi:10.3390/ijerph18178949_

Round 1
Reviewer 1 Report
The paper is relevant for the journal's aim (environmental research) given the strong focus on agriculture and landscape importance to protect from disasters, which also include plague (health dimension).
The introduction, though not concise, provides a good overview to what to expect from the paper. The paper follows what is promissed in the introduction.
The paper is not highly original, since there are plenty of studies on games about decision making in climate change (ex. in Springer), sessions at the European Geoscience Union General Assembly, papers by the group around Alenka Poplin in the US. Hence a better literature review in the introduction would be helpful. What gives the paper's originality is the link between urban planning and disaster risk reduction. This is indeed a topic not sufficiently explored and the serious game dimension even less. I recommend to include at least a reference to SimTorino, which simulates with serious games the development of Torino, building on SimCity. As known, SimCity also includes simulation of disasters.
The methods are well described, with a shortcoming. The shortcoming I compare to a game in Romania where the demolishers (and speculative builders) are likely to win over the restorers, because building restoration is expensive and requires much skill. How is this in this game? Are the fatalists (although not very chosen) likely to win over the ecologists? I mean, is it more game playing skill and experience required to win over the fatalists? This is not sufficiently clear from the questionnaire which rather evaluated the game playing experience, even if there is feedback on decision making. The way decision making is influenced by this simulation is well explained and thus the conclusions well supported by the discussion of the results.
The figures enrich well the paper and the supplementary material is useful and mainly translated.
I recommend to publish the paper, when addressing the discussed shortcomings.
Reviewer 2 Report
This is an interesting paper anchored on the following two objectives: (1) to present an education for sustainability (EfS) program and to evaluate its performance: a serious game of knowledge communication for the interactions among climate change, disaster risk, and urban development; (2) to explore factors that will influence the players’ decision-making in the trade-offs between urban development and DRR under an urbanization background through counterfactual scenarios constructed by the serious games. Despite the noted strengthens, just a couple of major observations around the research methodology, results, and conclusions that that would indeed enhance the quality of the revised manuscript.
Research methodology: A justification of the need of having a methodology with combined questionnaire survey and participant observation, complemented by interviews should be provided. Further, why was it necessary to have a total of 22 junior and senior undergraduates tested in the trial test? This explanation should be supported by relevant literature.
Likert scale – This is rather confusing around the labels attached with 5 as complete understanding and 1 indicated no knowledge at all. The emphasis should either be on levels of “understanding” OR “Knowledge” but not combined. Further, the questionnaire uses or rather adopts a five-point Likert scale to elicit opinions on the causes and impacts of variable climate hazards and the interaction between climate hazards and urbanization, and with twelve items were originally developed with reference to the cognitive learning objectives for climate action in Education for Sustainable Development Goals: Learning Objectives. However, no justification for the appropriateness of this scale is provided. It might be worthwhile to include within the questionnaire development section the reference to specific studies on causes and impacts of variable climate hazards regarding the adoption of the five-point Likert scale (as opposed to seven or nine). This approach would further provide some justification for the determination of the critical level or score for the causes and impacts of variable climate hazards (i.e. cut off point).
2.5 Analysis methods – Why was it necessary to have the quantitative data organized using Excel datasheets when the statistical analyses could easily be performed using SPSS?. What kind of t-test was performed, and why was this conducted? What was the t-test value used? This section is very weak and better explanations of the data analysis techniques employed supported by justifications with evidence or supporting references is necessary.
Results – The reporting of the results is very weak and this this needs to be cross referenced to the data within the Tables. For instance, the following reference to Table 4 (see Page 8) around “The t-test results indicate that the knowledge level of all of the students were significantly enhanced by the end of the serious game” doesn’t make sense as the authors fail to bring in the meaning of the significance values to supporting the findings. Further, some explanations of the three different groups (T1, T2 and T3) is desirable.
Interview data or results are reported but the actual analysis undertaken is not described.
Conclusions: The conclusion section needs to be revisited through inclusion of the emergent contributions to knowledge. Currently, this section is very narrow.
Round 2
Reviewer 2 Report
As indicated in the first round of review, this remains an interesting paper anchored on the following two objectives: (1) to present an education for sustainability (EfS) program and to evaluate its performance: a serious game of knowledge communication for the interactions among climate change, disaster risk, and urban development; (2) to explore factors that will influence the players’ decision-making in the trade-offs between urban development and DRR under an urbanization background through counterfactual scenarios constructed by the serious games. The authors have done an excellent job at addressing the reviewer’s comments around the research methodology, results, and conclusions are now fully addressed.